# From Artificial Needles to Real Haystacks: Improving Retrieval Capabilities in LLMs by Finetuning on Synthetic Data

**Zheyang Xiong**[w], **Vasilis Papageorgiou**[w], **Kangwook Lee**[w], **Dimitris Papailiopoulos**[w,ms]

[w]University of Wisconsin-Madison, [ms]Microsoft Research

## Abstract

Recent studies have shown that Large Language Models (LLMs) struggle to accurately retrieve information and maintain reasoning capabilities when processing long-context inputs. To address these limitations, we propose a finetuning approach utilizing a carefully designed synthetic dataset comprising numerical key-value retrieval tasks. Our experiments on models like GPT-3.5 Turbo and Mistral 7B demonstrate that finetuning LLMs on this dataset significantly improves LLMs' information retrieval and reasoning capabilities in longer-context settings. We present an analysis of the finetuned models, illustrating the transfer of skills from synthetic to real task evaluations (e.g., $10.5\%$ improvement on 20 documents MDQA at position 10 for GPT-3.5 Turbo). We also find that finetuned LLMs' performance on general benchmarks remains almost constant while LLMs finetuned on other baseline long-context augmentation data can encourage hallucination (e.g., on TriviaQA, Mistral 7B finetuned on our synthetic data cause no performance drop while other baseline data can cause a drop that ranges from $2.33\%$ to $6.19\%$). Our study highlights the potential of finetuning on synthetic data for improving the performance of LLMs on longer-context tasks.

## 1 Introduction

Recent studies have revealed that Large Language Models (LLMs) struggle to accurately retrieve information and maintain reasoning capabilities when processing longer context inputs or when retrieval is required across different parts of their context (Liu et al., 2023; Levy et al., 2024). These limitations hinder their performance on tasks that involve processing and reasoning over extensive textual information, such as summarization or question answering over long passages.

To address these challenges, we propose a novel approach that involves finetuning LLMs on a carefully designed fully numerical *synthetic* algorithmic dataset containing key-value dictionary retrieval tasks (*i.e.,* see Figure 1 for an example of such a task). We conduct extensive experiments on popular LLMs, including GPT-3.5 Turbo (OpenAI, 2023) and Mistral 7B (Jiang et al., 2023), and find that our method improves their performance on both information retrieval and long-context reasoning.

Specifically, our approach mitigates the "lost-in-the-middle" phenomenon identified by Liu et al. (2023) and significantly improves performance on the FLenQA benchmark (Levy et al., 2024) that measures LLMs' long-context reasoning capability. Interestingly, we observe that finetuning on our proposed dataset often yields more significant improvement compared to finetuning on the corresponding benchmark's data. In addition, it results in only a slight degradation on popular benchmarks such as MMLU (Hendrycks et al., 2021) and HellaSwag (Zellers et al., 2019), indicating that the overall capabilities of the models remain largely unaffected. Finally, another advantage of our proposed dataset is that it contains no *factual* information; as it was recently discovered by Gekhman et al. (2024), finetuning on previously unseen knowledge may encourage hallucinations.

---

Email: <zheyang@cs.wisc.edu>. Correspondence: <dimitris@papail.io>.

```
┌─────────────────── Simple dictionary key-value retrieval ───────────────────┐
│                                                                              │
│  Do a task using the list of dictionaries below.                            │
│                                                                              │
│  Dictionary [1] {122: 765, 4548: 1475, 4818: 4782}                          │
│  Dictionary [2] {526: 290, 9205: 9318, 9278: 1565}                          │
│  ...                                                                          │
│  Dictionary [32] {2931: 8364, 196: 1464, 812: 5363}                         │
│  ...                                                                          │
│  Dictionary [85] {344: 1579, 116: 617, 330: 411}                            │
│                                                                              │
│  Above is a list of dictionaries such that each key and value is an integer. │
│  Report the value of key 2931 and the dictionary it is in.                  │
│  ──────────────────────────────────────────────────────────────────────     │
│  Desired answer: The value of key 2931 is 8364 and it is in Dictionary [32]. │
│                                                                              │
└──────────────────────────────────────────────────────────────────────────────┘
```

Figure 1: An example prompt with desired answer of simple dictionary key-value retrieval task.

Thus, finetuning on our key-value dataset improves LLMs' retrieval and reasoning without suffering from such unwanted characteristics.

Our findings highlight the potential of finetuning on synthetic data as a promising approach to enhancing the performance of LLMs on real downstream tasks. Our paper is organized as follows: in Section 2 we describe the format of the proposed dataset, and its variations that provide (or not) an answer template to the model, in Section 3 we present our experimental results, in Section 4 we discuss the main limitations and possible future directions of our work, and in Section 5 we discuss our main conclusions.

## 1.1 RELATED WORK

**Long Context LLMs.** Recent works have observed LLMs' limited retrieval and reasoning capabilities in the long-context setting. Liu et al. (2023) discovered a positional bias when LLMs retrieve information from long contexts. In particular, the authors found out that the retrieval accuracy drops when the desired information lies in the middle of the context. Kamradt (2023) conducted the "needle-in-a-haystack" experiment by placing a random fact (the "needle") in a long input context (the "haystack") and observed that LLMs struggle to spot the needle as the input context length grows. To mitigate this behavior, Yu (2024) and An et al. (2024) finetuned LLMs on long-context augmentation data consisting of long-context question-answering tasks to enhance LLMs' long-context capabilities. Tang et al. (2023) shuffled the prompt and marginalized the prompt order biases in the long-context setting and Zhang et al. (2024) re-scaled the indices in positional encoding. Levy et al. (2024) introduced a benchmark, FLenQA, by extending input samples with varying lengths and types of padding, discovering LLMs' significant degradation in reasoning ability at context lengths much shorter than the maximum limit.

There are also other relevant works on long-context LLMs (Junqing et al., 2023; Mohtashami & Jaggi, 2023; Chen et al., 2023b; Bai et al., 2023; An et al., 2023). Xu et al. (2023) showed that Retrieval Augmented Generation (RAG) can be as accurate as full finetuning on longer context windows. Chen et al. (2023a) extended the LLM's predetermined context limit by treating it as an interactive agent who processes the input through iterative prompting. Jin et al. (2024) extended LLM's context window by remapping the unseen relative positions during inference. Zhu et al. (2024) introduced "LONGEMBED", a benchmark and suite of training-free strategies to extend embedding models' context window up to 32,768 tokens, leveraging Rotary Position Encoding (RoPE) in processing long contexts. Fu et al. (2024) proposed a data engineering recipe for scaling LLMs to $128k$ context lengths through lightweight continual pretraining on a balanced mixture of length-upsampled data. Peysakhovich & Lerer (2023) proposed "attention sorting," a method that improves long context models by iteratively sorting documents based on attention and generating responses with the re-ordered context.

```
┌─────────────── Multi-subkey dictionary key-value retrieval ───────────────┐

 Do a task using the list of dictionaries below.

 Dictionary [1] {(141, 986, 163): 2528, (726, 947, 349, 820): 4130}
 Dictionary [2] {(555, 710, 424): 5756, (623, 141, 997): 1633, (957, 634, 969): 7871}
 ...
 Dictionary [6] {(645, 417, 847): 6409, (141, 623, 616): 5617}
 ...
 Dictionary [49] {(710, 105, 141, 799): 5369, (623, 210, 477): 8971, (899, 126, 999):
 4409}

 Above is a list of dictionaries such that each key is a tuple of integers and each
 value is an integer. Report the key that contains the integers 616, 141, 623 (not
 necessarily in order), its value, and the dictionary it is in.
 ─────────────────────────────────────────────────────────────────────────
 Desired answer: The key that contains the integers 616, 141, 623 is (141, 623, 616).
 Its value is 5617 and it is in Dictionary [6].

└───────────────────────────────────────────────────────────────────────────┘
```

Figure 2: An example prompt with desired answer of multi-subkey dictionary key-value retrieval task. Here (141, 623, 616) is the *gold key*. Note that 141 and 623 in the *gold key* are also subkeys of other keys.

**Data-centric AI.**   In recent years, the field of data-centric AI has emerged, which focuses on improving the quality and efficiency of AI systems through data-oriented approaches rather than model-centric techniques (Sener & Savarese, 2018; Ghorbani & Zou, 2019; Zha et al., 2023; Albalak et al., 2024). Gadre et al. (2024) and Mazumder et al. (2024) proposed benchmarks that fix model training code, where the goal is to design better datasets to achieve better performance. Lee et al. (2023) and Zhou et al. (2024) studied the data format in training transformers to learn arithmetic tasks.

**LLM Benchmarks and Evals.**   Much research has been recently conducted towards the design of meaningful benchmarks that probe the capabilities of LLMs. Benchmarks such as GLUE (Wang et al., 2018), SuperGLUE (Wang et al., 2019) test whether a model has general language understanding capabilities. MMLU (Hendrycks et al., 2021) aims to measure the models' accuracy across a wide variety of tasks that span STEM, humanities, social sciences, and more, while GSM8k (Cobbe et al., 2021) tests capabilities on school math. In HellaSwag (Zellers et al., 2019) models are presented with an event description and must select the most likely follow-up sentence from a set of carefully selected choices, while HumanEval (Chen et al., 2021) measures their ability to generate code given docstrings. TriviaQA (Joshi et al., 2017) is a reading comprehension benchmark and NQ-Open (Lee et al., 2019; Kwiatkowski et al., 2019a) is an open domain question-answering benchmark where the question-answer pairs are collected from a diverse set of fields.

## 2 SYNTHETIC DATASET OF RETRIEVAL TASKS

In this section, we introduce the dataset on which we finetune the models. The dataset consists of two synthetic retrieval tasks: 1) simple dictionary key-value retrieval and 2) multi-subkey dictionary key-value retrieval.

**Simple dictionary key-value retrieval.**   In this task, we provide the model with a list of dictionaries of integer keys and values, and ask it to retrieve the value of a specified key (denoted as the *gold key*). Figure 1 shows an example of this task and the detailed algorithm is shown in Algorithm 2.

**Multi-subkey dictionary key-value retrieval.**   For models that can already tackle the first task (e.g., for the first task GPT 3.5 Turbo achieves around 0.99 accuracy irrespective of the position of *gold key*), we design a harder version of the key-value retrieval task where each key is a tuple of subkeys. Other keys can share some but not all of the subkeys of the *gold key*. We increase the difficulty of this

```
┌──────────────────────────────────────────────────────────────────────┐
│          Simple dictionary key-value retrieval (with an answer template)│
│                                                                          │
│  Do a task using the list of dictionaries below.                        │
│                                                                          │
│  Dictionary [1] {122: 765, 4548: 1475, 4818: 4782}                      │
│  Dictionary [2] {526: 290, 9205: 9318, 9278: 1565}                      │
│  ...                                                                      │
│  Dictionary [32] {2931: 8364, 196: 1464, 812: 5363}                     │
│  ...                                                                      │
│  Dictionary [85] {344: 1579, 116: 617, 330: 411}                        │
│                                                                          │
│  Above is a list of dictionaries such that each key and value is an integer. Report the │
│  value of key 2931 and the dictionary it is in. Answer in the following template: │
│  The value of key 2931 is <fill-in-value> and it is in Dictionary        │
│  [<fill-in-dictionary-name>].                                            │
│  ────────────────────────────────────────────────────────────────────  │
│  Desired answer: The value of key 2931 is 8364 and it is in Dictionary [32]. │
└──────────────────────────────────────────────────────────────────────┘
```

Figure 3: The prompt of the simple dictionary key-value retrieval task is provided with an answer template.

Instruction
```
... Report the value of key 2931 and the
dictionary it is in.
```
Target Answer
```
The value of key 2931 is 8364 and it is
in Dictionary [32].
```

Instruction
```
... Report the value of key 2931 and the
dictionary it is in. Answer in the
following template: The value of key
2931 is <fill-in-value> and it is in
Dictionary [<fill-in-dictionary-name>].
```
Target Answer
```
The value of key 2931 is 8364 and it is
in Dictionary [32].
```

Figure 4: Token-level loss on the target answer when provided with (right) and without (left) an answer template, where red indicates high and green low loss.

task by randomizing the order of subkeys in the prompt so that the order is not necessarily the same as that of the *gold key*. Figure 2 shows an example of this task and the detailed algorithm is shown in Algorithm 3.

**Prompt with an answer template.** Note that with the prompt in Figure 1, slightly different answers like "8364 is the value of key 2931 in dictionary 32" and "Dictionary [32] has the key 2931 with value 8364" are also correct. Therefore, since the model is finetuned on the entire answer, during supervised finetuning, it also learns the format of our provided answer besides learning to retrieve the desired value. In order to make the model only focus on retrieving the correct value without being affected by the format of the answer, we provide the model with an answer template with which we want the model to answer. Figure 3 shows an example of a prompt with an answer template. In Figure 4 we visualize the token-level loss on the target answer, where red indicates high and green low loss. If an answer template is provided, the loss on the formatting part is small. This lets the model to focus on the important part and learn the right skill rather than how to answer the question.

## 3 EXPERIMENTS AND RESULTS

Our goal is to investigate whether finetuning LLMs (in particular, GPT-3.5 Turbo and Mistral 7B [1]) on our proposed synthetic numerical retrieval tasks improves their long context capabilities on

---
[1] `gpt-3.5-turbo-1106` and `Mistral-7B-Instruct-v0.1`

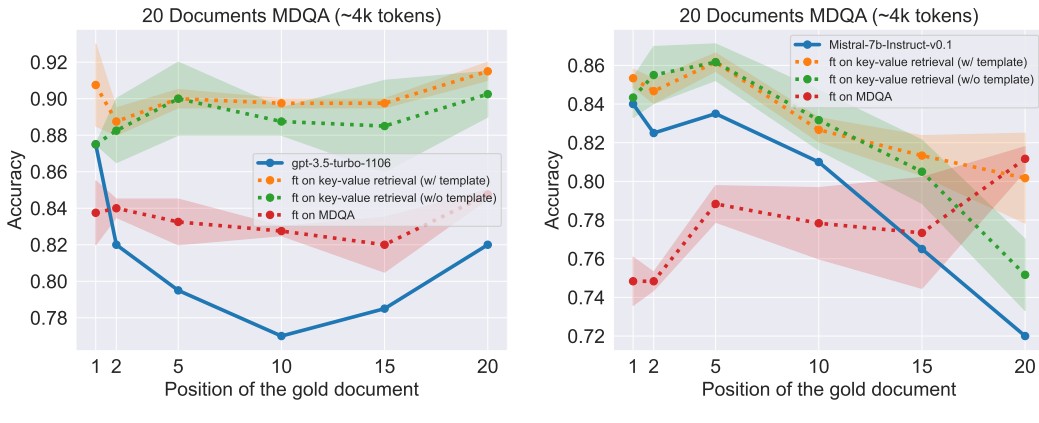

(a) GPT-3.5 Turbo and the finetuned versions.  (b) Mistral 7B and the finetuned versions.

Figure 5: Performance of GPT-3.5 Turbo, Mistral 7B and their corresponding finetuned versions on the MDQA task.

natural language tasks: multi-document question answering (MDQA) (Liu et al., 2023) and flexible length question answering (FLenQA) (Levy et al., 2024).

### 3.1 STAGE 1: FINETUNING LLMS ON SYNTHETIC RETRIEVAL TASKS

For Mistral 7B, our dataset consists of 350 samples of simple dictionary key-value retrieval tasks. Each task has 85 dictionaries and each dictionary has 3 to 4 keys, so each prompt has roughly 4K tokens. We finetune the model on only the answer part (masking out the instruction part) for 2 epochs. More implementation details are in A.1. Figure 11 shows Mistral 7B's performance on simple dictionary key-value retrieval task before and after finetuning.

Since GPT-3.5 Turbo already performs well on simple dictionary key-value retrieval task, we finetune it on multi-subkey dictionary key-value retrieval tasks. The dataset consists of 150 samples and each sample has 49 dictionaries. We finetune the model for 3 epochs using OpenAI's API.

### 3.2 STAGE 2: EVALUATIONS ON LONG CONTEXT RETRIEVAL AND REASONING TASKS

#### 3.2.1 MULTI-DOCUMENT QUESTION ANSWERING (MDQA)

We test models' capabilities of retrieving important information in a long context setting. In MDQA, we provide the model with $k$ documents and prompt it to answer a question such that only 1 of $k$ documents (denoted as the *gold document*) contains the answer and the other $k-1$ documents (denoted as *distractors*) are completely irrelevant to the question. We test the setting of a context with 20 documents (around 4K tokens) and place *gold document* at positions $\{1, 2, 5, 10, 15, 20\}$ [2]. For each position, we test the model on 200 task samples and measure the accuracy using the maximum subspan exact match as in (Liu et al., 2023).

> **Finding 1:** *Finetuning LLMs on synthetic key-value retrieval tasks enhances their performance on practical retrieval tasks, demonstrating effective transfer of learned capabilities.*

The result of 20 documents MDQA is shown in Figure 5, where x-axis is the position of *gold document*. In Figure 5a, for the original GPT-3.5 Turbo model, there is a U-shaped performance curve, indicating that the performance is highest if the important information is at the beginning or at the end of the input context, with the model struggling to retrieve the answer if the important information is in the middle. Finetuning the models on synthetic retrieval tasks flattens the U-shaped curve and information is much more accurately retrieved over all positions across the input context. In Figure 5b, the original Mistral 7B model has a primacy bias – in the sense that it can more accurately

---

[2]For example, *gold document* placed at position 1 means it is the first document in the context.

retrieve information that is at the beginning of the input context. Finetuning the models on our proposed data manages to improve the accuracy across all the positions in the input context. In addition, when the finetuning dataset contains a template, Mistral seems to mitigate this primacy bias, showcasing a more uniform accuracy across all the positions in the input context.

> **Finding 2:** *Synthetic data is better than MDQA data even if the goal is to perform better in MDQA task.*

As a comparison, we also finetune the models on the MDQA dataset itself for roughly the same number of training tokens and see how finetuned models perform. Since the MDQA dataset only provides the ground truth answers in one or two words, we prompt GPT-3.5 Turbo with correct answers and let it form a complete sentence as the target answer. As shown in Figure 5a, GPT-3.5 Turbo finetuned on our synthetic data perform better than the one finetuned on MDQA. In Figure 5b we can see that despite training on MDQA tasks, Mistral 7B still struggles to perform well on MDQA, with a significant performance drop when *gold document* is at the beginning of the prompt. These findings underscore the effectiveness of our synthetic data generation method, which enhances performance on specific datasets like MDQA, even surpassing direct finetuning on the target dataset.

### 3.2.2 FLEXIBLE LENGTH QUESTION ANSWERING (FLENQA)

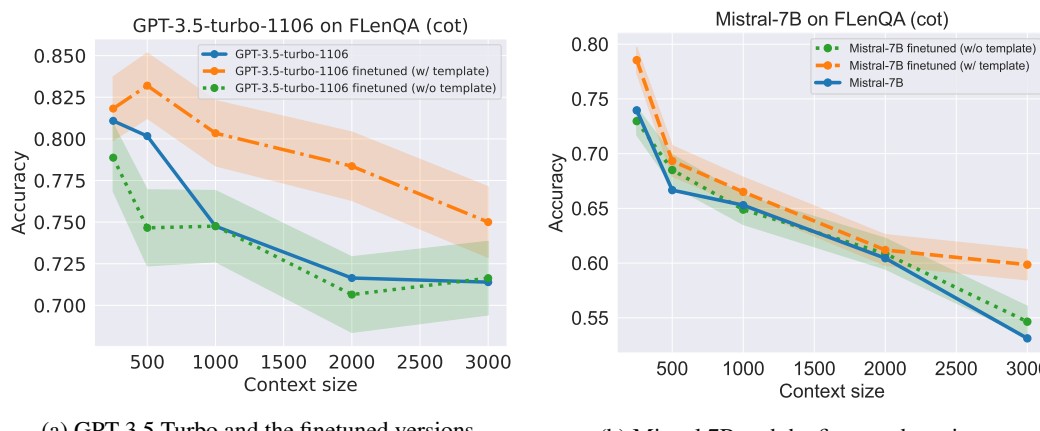

(a) GPT-3.5 Turbo and the finetuned versions.

(b) Mistral 7B and the finetuned versions.

Figure 6: Performance of GPT-3.5 Turbo, Mistral 7B and their corresponding finetuned versions on the FLenQA task, using chain-of-thought prompting.

We also test models' long context reasoning capabilities. FLenQA is a dataset comprising reasoning tasks with varying length that ranges from 250 tokens to 3000 tokens. Each task consists of a context and a "True" or "False" question that can be answered by two key sentences from the context. We test chain-of-thought (Wei et al., 2022) and non chain-of-thought prompting, each with a total of 2000 task samples. For chain-of-thought prompting, we ask the model to produce the result step by step and derive the answer ("True" or "False") at the end, and in the non chain-of-thought prompting we ask the model to directly answer "True" or "False".

> **Finding 3:** *Finetuning LLMs on synthetic key-value retrieval tasks improves LLMs' long-context reasoning capabilities, even if explicit chain-of-thought reasoning is not allowed.*

In Figure 6 and 7 we present our results on the FLenQA dataset. The x-axes represent the number of tokens in the context, while the y-axes represent the accuracy of the response. Figure 6 shows results where chain-of-thought prompting is employed. In Figure 6a, we notice that although the model suffers from a performance drop if finetuned on data without answer template, finetuning GPT-3.5 Turbo on data with answer template significantly improves model's chain-of-thought reasoning capability. In Figure 6b we can also see that finetuning Mistral 7B on data with answer template improves models chain-of-thought capability. We hypothesize that the reason for this is that the

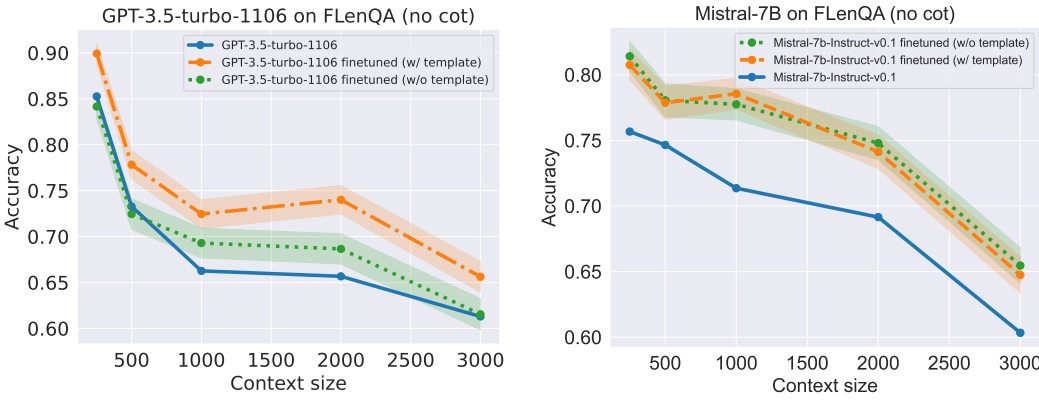

(a) GPT-3.5 Turbo and the finetuned versions.  (b) Mistral 7B and the finetuned versions.

Figure 7: Performance of GPT-3.5 Turbo, Mistral 7B and their corresponding finetuned models on the FLenQA task without employing chain-of-thought prompting.

finetuned models utilize their improved retrieval capabilities to capture relevant information more accurately, which helps them deduce the answer.

Figure 7 presents results where models are required to directly answer with "True" or "False" without providing explicit reasoning. The results show a notable improvement in performance for finetuned models. This improvement is significant because it demonstrates that, even if explicit reasoning (that is related to retrieval capability) is not allowed, finetuning on our proposed synthetic tasks enhances the models' internal reasoning capabilities.

> **Finding 4:** *LLMs finetuned on synthetic tasks with answer templates are better.*

From Figure 5, 6 and 7, we can observe that models finetuned on synthetic key-value retrieval tasks with answer templates perform better on MDQA and FLenQA than that on without answer templates. This verifies our hypothesis that having an answer template helps the model learn the right skill more efficiently. This highlights a key advantage of synthetic data: it allows for greater control over the model's output format. Unlike real-world tasks where developing answer templates can be challenging, synthetic tasks allow for easy implementation of structured response formats, facilitating skill learning.

### 3.3 STAGE 3: EVALUATION OF FINETUNED MODELS' GENERAL CAPABILITIES

> **Finding 5:** *Finetuning LLMs on synthetic key-value retrieval tasks does not hurt models' general capabilities.*

One possible drawback of our approach is that finetuning on the proposed artificial tasks would severely harm the general purpose capabilities of the tested models. In order to assess this concern, we tested the original and finetuned versions of GPT-3.5 Turbo and Mistral 7B on some general purpose benchmarks. Note that for our assessments we used the codebases of Gao et al. (2023) and Fu et al. (2023).

The results can be seen in Table 1. In particular, we consider five widely used benchmarks: MMLU (Hendrycks et al., 2021)[3], HellaSwag (Zellers et al., 2019), GSM8k (Cobbe et al., 2021), TriviaQA (Joshi et al., 2017) and NQ-Open (Kwiatkowski et al., 2019b). What we can observe is that all the finetuning strategies result in no significant degradation on the general purpose benchmarks mentioned above.

---

[3]Due to computational constraints, we did not evaluate GPT-3.5 Turbo on all benchmarks, and for MMLU we use 20% of the full dataset.

Table 1: Model's performance evaluated on general ability benchmarks. All numbers are reported in percentage. Here "w/" and "w/o" denote the models that are finetuned on the the synthetic tasks that were described in Section 2.

| MODEL | MMLU | HellaSwag | GSM8K | Triviaqa | NQ-Open |
|---|---|---|---|---|---|
| Mistral-7B | 53.42 | 56.31 | 34.65 | 47.63 | 11.61 |
| Mistral-7B ft (w/template) | 53.44 (+0.02) | 56.22 (−0.09) | 34.34 (−0.31) | 47.74 (+0.11) | 11.98 (+0.37) |
| Mistral-7B ft (w/o template) | 53.42 (−0.00) | 56.30 (−0.01) | 34.14 (−0.51) | 47.62 (−0.01) | 11.40 (−0.21) |
| GPT-3.5-turbo | 68.07 | - | 72.33 | - | - |
| GPT-3.5-turbo ft (w/template) | 67.75 (−0.32) | - | 71.65 (−0.68) | - | - |
| GPT-3.5-turbo ft (w/o template) | 68.16 (+0.09) | - | 75.06 (+2.73) | - | - |

## 3.4 STAGE 4: COMPARISONS WITH OTHER BASELINES

We also consider three additional long-context augmentation datasets as baselines: MultidocQA (Yu, 2024), IN2 (An et al., 2024), and Needle-in-a-haystack (Kamradt, 2023). MultidocQA is a dataset of multiple documents question and answering where the model needs to paraphrase the document before answering. IN2 is a long-context question answering dataset where the answer can be deduced from one or multiple parts of the context. Needle-in-a-haystack is a widely used long-context test set where the model is prompted to identify some key information (the needle) within a long context (the haystack). We finetune Mistral 7B on these baselines, using roughly the same number of training tokens and report their performance on MDQA, FLenQA, and general purpose benchmarks.

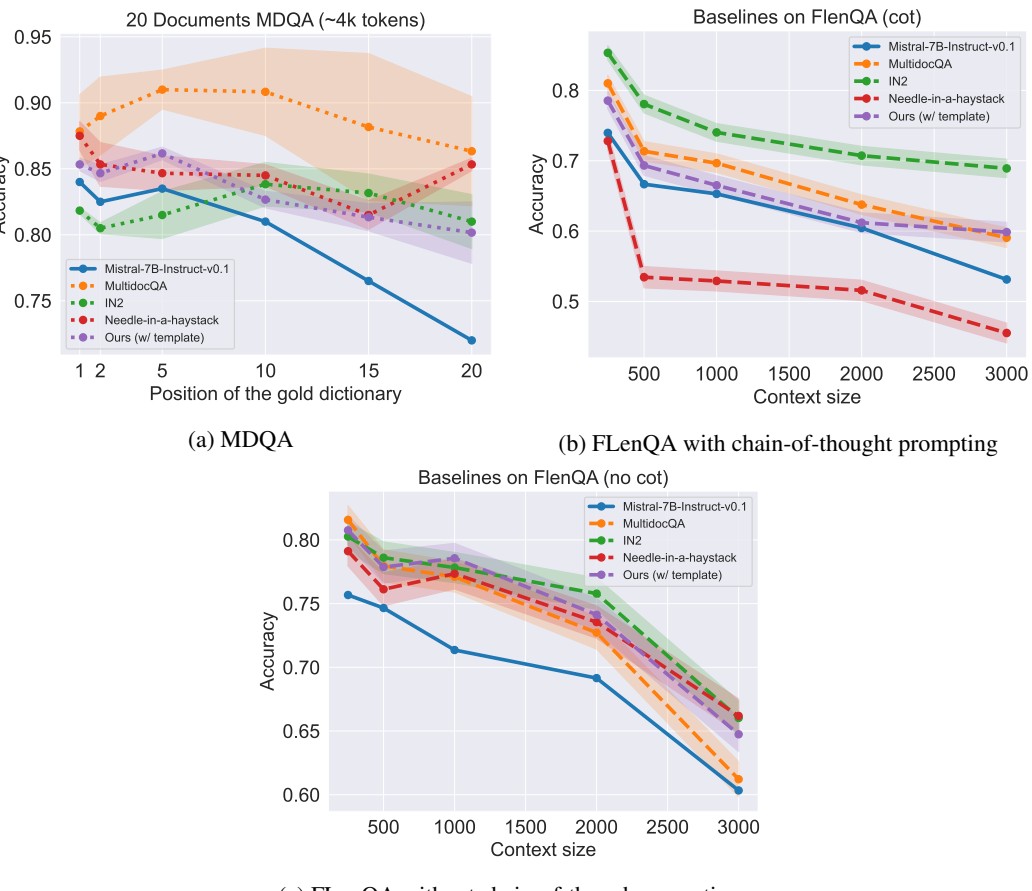

(a) MDQA

(b) FLenQA with chain-of-thought prompting

(c) FLenQA without chain-of-though prompting

Figure 8: Performance of finetuned Mistral 7B on (a) MDQA, (b) FLenQA with chain-of-thought prompting, and (c) FLenQA without chain-of-thought prompting.

Table 2: Mistral 7B and finetuned versions' performance evaluated on general ability benchmarks. All numbers are reported in percentage.

| Finetuning dataset | MMLU | HellaSwag | GSM8K | Triviaqa | NQ-Open |
|---|---|---|---|---|---|
| Original Mistral-7B | 53.42 | 56.31 | 34.65 | 47.63 | 11.61 |
| Ours (w/template) | 53.44 (+0.02) | 56.22 (−0.09) | 34.34 (−0.31) | 47.74 (+0.11) | 11.98 (+0.37) |
| MultidocQA (Yu, 2024) | 53.19 (-0.22) | 56.27 (-0.04) | 33.28 (-1.36) | 45.20 (-2.43) | 8.69 (-2.91) |
| IN2 (An et al., 2024) | 53.49 (+0.07) | 56.44 (+0.13) | 34.98 (+0.32) | 45.44 (-2.19) | 9.80 (-1.81) |
| Needle-in-a-haystack (Kamradt, 2023) | 52.83 (-0.59) | 56.22 (-0.09) | 33.79 (-0.86) | 41.30 (-6.33) | 4.88 (-6.73) |
| MDQA (Liu et al., 2023) | 52.94 (-0.47) | 56.23 (-0.07) | 34.72 (-0.07) | 44.77 (-2.85) | 7.64 (-3.96) |

**Finding 6:** *Synthetic data do not encourage hallucinations that other baselines may yield.*

From Figure 8 and Table 2, we can see that while some baselines outperform our proposed data on either MDQA or FLenQA, they all have more significant degradation on the general benchmarks we test, especially on TriviaQA and NQ-Open. One possible reason is that all other baselines contain factual information. Gekhman et al. (2024) shows that finetuning on factual information encourages hallucinations, something that we verify observing the significant degradation on TriviaQA and NQ-Open, which are knowledge-based benchmarks. In contrast, our proposed dataset is purely synthetic, comprising of key-value pairs, and as a result, does not encourage hallucinations. We also highlight another benefit of our synthetic data: since it does not contain any factual information, it will not have the problem of containing potential outdated information that further encourages hallucinations, from which other long-context augmentation datasets may suffer.

## 3.5 STAGE 5: EVALUATION ON LONGER-CONTEXT SETTING

We also test the longer-context setting. We finetune `Mistral-7b-Instruct-v0.2` on simple key-value retrieval task with maximum context length of 24K and test it on MDQA. We observe a clear improvement over the original model as shown in Figure 9 and does not observe significant degradation in general capability, shown in Table 3.

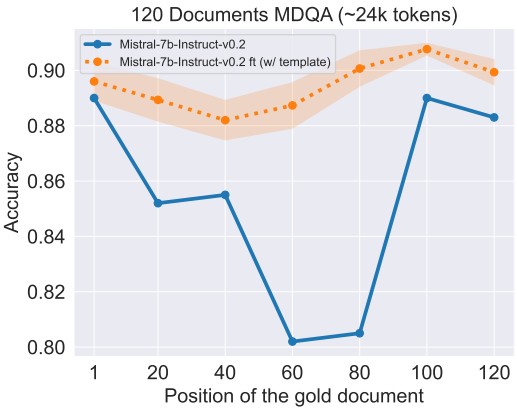

Figure 9: Performance of finetuned `Mistral-7b-Instruct-v0.2` on 120 documents MDQA.

Table 3: The performance of Mistral-7B-Instruct-v0.2 and finetuned version (on simple dictionary key-value retrieval with context length of 24K) on general capability benchmarks. All numbers are reported in percentage.

| Finetuning dataset | MMLU | HellaSwag | GSM8K | TriviaQA | NQ-Open |
|---|---|---|---|---|---|
| Mistral-7B-Instruct-v0.2 | 58.74 | 65.98 | 41.85 | 32.65 | 14.46 |
| Ours (w/ template) | 58.42 (−0.32) | 65.74 (−0.24) | 41.72 (−0.13) | 33.02 (+0.37) | 15.03 (+0.57) |

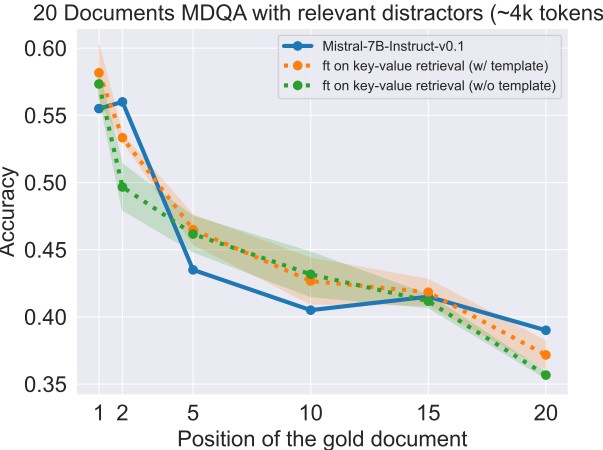

Figure 10: Mistral 7B and the finetuned versions on MDQA with relevant distractors. The finetuned variants do not show a significant improvement over the original model.

## 4    LIMITATIONS AND FUTURE WORK

Our dataset does have a limitation. MDQA benchmark also has another version where *distractors* are relevant distractors, meaning that they are documents retrieved by a retrieval system (based on the relevance score) that do not contain the answer. Models finetuned on our dataset will not improve in this setting, as is shown in Figure 10. A possible future work of this study is to add our synthetic retrieval dataset as a small part of a larger instruction finetuning dataset and see the difference between models finetuned with and without synthetic retrieval data and observe how they perform differently on long context retrieval and reasoning tasks.

## 5    CONCLUSION

In this work, we introduce a novel finetuning approach that leverages carefully designed synthetic datasets to enhance the information retrieval and reasoning capabilities of LLMs in real downstream tasks. Our study demonstrates that finetuning on our proposed synthetic data significantly improves the performance of the tested models on tasks like MDQA and FLenQA, mitigating the "lost-in-the-middle" behavior that was observed in Liu et al. (2023). On the other hand, we find that after finetuning, the models' performance on general benchmarks remains almost constant, something that indicates that their overall capabilities are mostly unaffected. We also find that compared to other long-context augmentation datasets that contain factual information, our purely artificial data does not encourage hallucinations. Moreover, it will not have the problem of containing potential outdated information. Thus, we believe that our study demonstrates the potential of finetuning LLMs on carefully crafted synthetic datasets to enhance their capabilities on downstream tasks. We hope that our findings will inspire further research into the development of effective synthetic datasets.

### ACKNOWLEDGEMENT

This work was supported by NSF CAREER Award CCF2339978, an Amazon Research Award, a grant from FuriosaAI, and ONR Grant No. N00014-21-1-2806 and No. N00014-23-1-2848.

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

## A EXPERIMENT DETAILS

### A.1 FINETUNING MISTRAL 7B AND GPT 3.5 TURBO

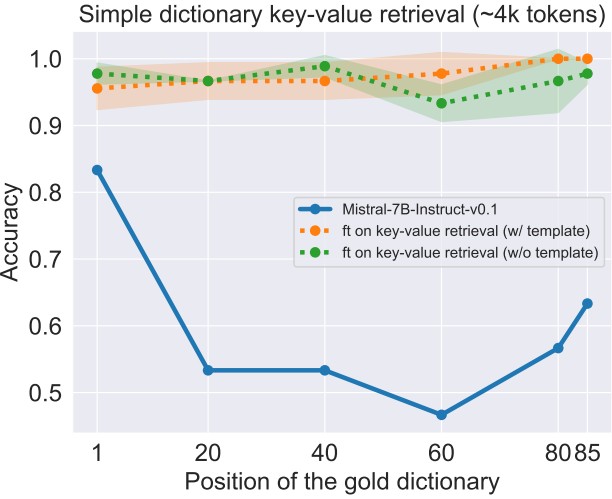

Figure 11: Mistral 7B and the finetuned versions on simple dictionary key-value retrieval.

For Mistral 7B, we choose simple dictionary key-value retrieval as the task to finetune on. We use two prompting strategies to prepare the dataset: with and without an answer template as described in Section 2. For each prompting strategy we generate 3 different datasets using the same configuration but with different seeds. Each dataset consists of 350 simple dictionary key-value retrieval tasks (roughly 4K tokens in each task). Each task has 85 dictionaries and each dictionary has 3 to 4 keys. Each key and value is an integer of 3 to 4 digits (in particular, we choose $l_{min} = r_{min} = 3, l_{max} = r_{max} = 4$). We finetune Mistral 7B on all attention layers and use a global batch size of 16 and finetune the model for 2 epochs on each dataset with learning rate $5 \times 10^{-6}$. For evaluation results, we average across 3 runs, each with different training data and seed.

For GPT-3.5 Turbo, we choose multi-subkey key-value retrieval as the task to finetune on (in particular, we choose `num_dict` $= 49, l_{min} = r_{min} = 3, l_{max} = r_{max} = 4,$ `n_keys` $= 3,$ `n_common` $= 2.p_{share} = 0.5$). For each prompting strategy, we generate 2 different datasets. Each dataset consists of 150 multi-subkey key-value retrieval tasks (roughly 4K tokens in each task). Each task has 49 dictionaries. We finetune GPT-3.5 Turbo for 2 epochs on each dataset using OpenAI API. For evaluation results, we average across 2 runs.

### A.2 EVALUATION DETAILS

We use lm-eval-harness (Gao et al., 2023) to for evaluation and use the default configuration. In particular, for GSM8K we use 5-shot prompting and for other tasks we use 0 shot prompting. An exception is for `Mistral-7b-Instruct-v0.2`, where we use 1 shot prompting for NQ-Open as the model cannot well answer in the desired format in 0 shot prompting setting.

# B  DETAILS ON GENERATING RETRIEVAL TASKS

In this section we provide the pseudocodes on generating retrieval tasks introduced in the paper: (1) simple dictionary key-value retrieval and (2) multi-subkey dictionary key-value retrieval.

## B.1  SIMPLE DICTIONARY KEY-VALUE RETRIEVAL

---

**Algorithm 1:** Gen_key_val

---

**Input:** min and max number of digits of key / value $r_{min}, r_{max}$, gold key gold_key
**Output:** key and val where key is different from gold_key

1  val $\leftarrow$ randint($r_{min}, r_{max}$)
2  **while** *True* **do**
3  |    key $\leftarrow$ randint($r_{min}, r_{max}$)
4  |    **if** key $!=$ gold_key **then return** key, val

---

**Algorithm 2:** Simple dictionary key-value retrieval

---

**Input:** Number of dictionaries num_dict; min and max length of each dictionary $l_{min}, l_{max}$; range of all keys / values ($r_{min}, r_{max}$)
**Output:** A list of dictionaries dicts, the position of gold dictionary gold_pos, gold key gold_key and gold value gold_val.

1  Initialize gold_dict as an empty dictionary
2  gold_dict_len $\leftarrow$ randint($l_{min}, l_{max}$)
3  gold_pos $\leftarrow$ randint(1, num_dict)
4  gold_key $\leftarrow$ randint($r_{min}, r_{max}$)
5  gold_val $\leftarrow$ randint($r_{min}, r_{max}$)
6  Add (gold_key, gold_val) key-value pair to gold_dict
7  **for** $i = 1, \ldots, gold\_dict\_len - 1$ **do**
8  |    key, val $\leftarrow$ Gen_key_val($r_{min}, r_{max}$, gold_key)
9  |    Add (key, val) key-value pair to gold_dict
10  Shuffle the order of gold_dict.
11  Initialize dicts to an empty array of dictionaries
12  **for** $j = 1, \ldots, num\_dict - 1$ **do**
13  |    Initialize dict as an empty dictionary
14  |    dict_len $\leftarrow$ randint($l_{min}, l_{max}$)
15  |    **for** $k = 1, \ldots, dict\_len$ **do**
16  |    |    key, val $\leftarrow$ Gen_key_val($r_{min}, r_{max}$, gold_key)
17  |    |    Add (key, val) key-value pair to dict
18  |    Append dict to dicts
19  Insert gold_dict into dicts at position gold_pos
20  **return** dicts

---

## B.2 Multi-subkey key-value retrieval

---

**Algorithm 3:** Gen_multikey_val

---

**Input:** range for all keys / values: $(r_{min}, r_{max})$, gold multi-key: gold_key_tuple, number of keys in each multi-key: n_keys, keys from gold_key_tuple that can be shared with the output key_tuple: common_subkey, probability of key sharing: $p_{\text{share}}$

**Output:** key_tuple and corresponding val

1  **assert** len(common_subkey) < n_keys
2  val $\leftarrow$ randint$(r_{min}, r_{max})$
3  **while** *True* **do**
4      key$_i \leftarrow$ randint$(r_{min}, r_{max})$, $\forall i = 1, 2, \ldots,$ n_keys
5      key_tuple $= ($key$_1,$ key$_2, \ldots,$ key$_{\text{n\_keys}})$
6      **for** $i = 1, ...,$ *len(common_subkey)* **do**
7         With probability $p_{\text{share}}$ replace key$_i$ with common_subkey$_i$.
8      Shuffle the elements of key_tuple.
9      **if** *key_tuple and gold_key_tuple share at most len(common_subkey) keys* **then**
10        **return** key_tuple, val

---

**Algorithm 4:** Multi-subkey dictionary retrieval

---

**Input:** Number of dictionaries: num_dict, min and max length of each dictionary: $l_{min}, l_{max}$, range of each key / value: $(r_{min}, r_{max})$, number of keys in each multikey: n_keys, max number of keys to share among key_typle's: n_common, probability of key sharing between keys: $p_{\text{share}}$.

**Output:** A list of dictionaries dicts, the position of gold dictionary gold_pos, gold multi-key gold_key_tuple and gold value gold_val.

1  **Assert** n_common < n_keys.
2  Initialize gold_dict as an empty dictionary
3  gold_dict_len $\leftarrow$ randint$(l_{min}, l_{max})$
4  gold_pos $\leftarrow$ randint$(1,$ num_dict$)$
5  gold_key$_i =$ randint$(r_{min}, r_{max})$, $\forall i = 1, 2, \ldots,$ n_keys
6  gold_key_tuple $= ($gold_key$_1,$ gold_key$_2, \ldots,$ gold_key$_{\text{n\_keys}})$
7  gold_val $\leftarrow$ randint$(r_{min}, r_{max})$
8  Choose n_common random keys from gold_key_tuple.
9  Add (gold_key_tuple, gold_val) key-value pair to gold_dict
10 **for** $i = 1, \ldots,$ *gold_dict_len* $- 1$ **do**
11     key_tuple, val $\leftarrow$
       Gen_multikey_val$(r_{min}, r_{max},$ gold_key_tuple, n_keys, $p_{\text{share}})$.
12     Add (key_tuple, val) multikey-value pair to gold_dict
13 Shuffle the order of gold_dict.
14 Initialize dicts to an empty list.
15 **for** $j = 1, \ldots,$ *num_dict* $- 1$ **do**
16     Initialize dict as an empty dictionary
17     dict_len $\leftarrow$ randint$(l_{min}, l_{max})$
18     **for** $k = 1, \ldots,$ *dict_len* **do**
19        key_tuple, val $\leftarrow$ Gen_multikey_val$(r_{min}, r_{max},$ gold_key$)$
20        Add (key_tuple, val) multikey-value pair to dict
21     Append dict to dicts
22 Insert gold_dict into dicts at position gold_pos
23 **return** dicts

---

