# OpenReview forum: "From Artificial Needles to Real Haystacks: Improving Retrieval Capabilities in LLMs by Finetuning on Synthetic Data"
_ICLR.cc/2025/Conference — ICLR 2025 Poster_

### Official Review · Reviewer_19vD · 2024-10-30

**Soundness:** 3
**Presentation:** 3
**Contribution:** 3
**Rating:** 6
**Confidence:** 3

**Summary:**

The paper proposes synthetic data generating for the tasks requiring  understanding of long context. Specifically, it generates a key-value retrieval task where an LLM is fine-tuned to find out a dictionary with the specified key value. To make the task more difficult,  the key becomes a tuple of integers and the order of integers are randomly shuffled. After fine-tuning the LLM, LLM is evaluated on MDQA and FlenQA dataset. The proposed synthetic data generation leads to performance improvement of  GPT-3.5-Turbo and Mistral.

**Strengths:**

- The proposed method is simple and effective.

- It does not degrade the performance on other benchmark datasets such as MMLU, HellaSwag, GSM8K, Triviaqa, NQ-open.

- The paper is well written.

**Weaknesses:**

- Details of how key-value retrieval task is generated are missing, which seems to be critical.

- More extensive experiments need for validating the proposed method. For example, I highly recommend to evaluate the method on benchmark datasets such as RULER [1] and Long Bench [2] with different backbone LLMs such as Llama and Gemma.



## References
[1] Hsieh, Cheng-Ping, et al. "RULER: What's the Real Context Size of Your Long-Context Language Models?." arXiv preprint arXiv:2404.06654 (2024).

[2] Bai, Yushi, et al. "Longbench: A bilingual, multitask benchmark for long context understanding." arXiv preprint arXiv:2308.14508 (2023).

**Questions:**

- Did you full fine-tune the Mistral or use parameter efficient fine-tuning (peft) such as lora?

- What is the rationale for choosing the number of samples in key-value retrieval tasks?

- In Figure 5b, Mistral-v0.1 is used, while Mistral-v0.2 is used in Figure 7b. Is there any reason why you use different version of Mistral?

---

> ### Author Response · Authors · 2024-11-30
> **Response to Reviewer 19vD (1/2)**
>
> Dear Reviewer 19vD,
>
> We would first like to apologize for our delayed reply. We greatly appreciate your constructive feedback on our paper. Below, we address the specific concerns raised in the review.
>
> **W1: Details of retrieval tasks are missing**
>
> We thank the reviewer for pointing this out. In our Appendix C of our updated draft (page 21-23), we include detailed algorithm implementations on how to generate our synthetic data. We also update Appendix A.1 (page 13) to include more details. We will also open source our code for generating synthetic data.
>
> **W2: Evaluation on RULER and LongBench**
>
> We thank the reviewer for the suggestion and we plan to add more discussion on RULER and LongBench. We would first like to explain why we do not evaluate on RULER [3]. RULER is a synthetic evaluation task that includes "dictionary key-value" like retrieval task, which is similar to our task. For example, RULER has single key-value retrieval, multi-keys retrieval and multi-values retrieval, where they explicitly use integers as values to retrieve, which is similar to our dictionary key-value retrieval tasks. Our work focuses on synthetic to real generalization, where we investigate whether finetuning on synthetic (symbolic) task improves model's retrieval capabilities in real task. Therefore, testing model's capability on other synthetic (symbolic) long-context tasks is beyond the scope of our work. A possible direction for future work is to finetune on different synthetic tasks introduced in RULER and whether the model's long-context capabilities improce in the real settings.
>
> We evaluate on LongBench [4], which has various real tasks, and we present the result in [Table II](https://github.com/PlJQ/needles/blob/main/Table_II.pdf).
>
> While we observe that all models have some degradation, we would like to argue that the message is inconclusive here and LongBench is not suitable for the setting that we study for two reasons.
> 1. LongBench is not very reliable for small-sized model like 7B. We notice from [Table II](https://github.com/PlJQ/needles/blob/main/Table_II.pdf) that the performance decreases, and this is likely due that LongBench mostly uses F1 and Rouge-L scores in the evaluation where the answer length matters, and sometimes answers that are equivalently correct can have different scores when evaluated on LongBench. For example, if the question is `What is the decoder`, the reference answer is `LSTM decoder` and the predicted answer is `The decoder is an LSTM decoder`, while the predicted answer is correct, the F1 score would be 0.5. While [4] uses prompting strategy to make LLMs produce shorter answer, the prompting strategy is not always reliable for model with smaller size like 7B. The drop in [Table II](https://github.com/PlJQ/needles/blob/main/Table_II.pdf) can be caused by that different long-context augmentation data changes the "style" of model's answer and the model might generate longer but equivalently correct answer. On the other hand, the tests introduce in our paper are accuracy-based and more reliable.
> 2. Our setting is different from previous works that evaluate on LongBench. Our work considers finetuning on instruction-tuned model with a small size of data. In contrast, previous works like [5] instruction-tune a base model using a large long-context augmentation dataset (training dataset size of 1.4 million) and a general instruction-tuning data of size 200K and observe an improvement on LongBench. A corresponding setting for our work would be to instruction-tune on a large set of synthetic data and see if the performance increases.
>
> We will include this discussion in our revised manuscript. We are happy to discuss if you have further concerns on this.

---

> ### Author Response · Authors · 2024-11-30
> **Response to Reviewer 19vD (2/2)**
>
> **Q1: Did you full fine-tune the Mistral or use parameter efficient fine-tuning (peft) such as lora?**
>
> We finetune Mistral directly on the weight matrices and do not use LoRA.
>
> **Q2: Rationale for choosing the number of samples**
>
> There is no specific rationale on how we choose the number of samples. We select relatively small numbers as the size of training set and show that training the model on this small dataset improve model's performance on MDQA and FLenQA.
>
> In Appendix B.1 of our updated draft, we conduct additional ablation studies to investigate how the amount of training affect the model's performance.
>
> **Q3: In Figure 5b, Mistral-v0.1 is used, while Mistral-v0.2 is used in Figure 7b. Is there any reason why you use different version of Mistral?**
>
> > while Mistral-v0.2 is used in Figure 7b
>
> We think you might mean "Figure 9" here as we use Mistral-v0.1 in Figure 7b. We apologize for causing the confusion. Most our experiments are in the 4K setting as the original "lost-in-the-middle" paper [1] considers this setting and FLenQA [2] has maximum context size 3K. Context length 4K is also more convenient for us to run more experiments on evaluations due to computational constraints and therefore chose Mistral-v0.1 for most of our experiments. We decide to also test the longer-context setting (24K) to see if it improves on MDQA and therefore choose Mistral-v0.2 as it supports longer context length.
>
> We thank again for your constructive feedback and would like to discuss any remaining questions.
>
> **References**
>
> [1] Liu, N. F., Lin, K., Hewitt, J., Paranjape, A., Bevilacqua, M., Petroni, F., & Liang, P. (2024). Lost in the middle: How language models use long contexts. Transactions of the Association for Computational Linguistics, 12, 157-173.
>
> [2] Levy, M., Jacoby, A., & Goldberg, Y. (2024). Same task, more tokens: the impact of input length on the reasoning performance of large language models. arXiv preprint arXiv:2402.14848.
>
> [3] Hsieh, Cheng-Ping, et al. "RULER: What's the Real Context Size of Your Long-Context Language Models?." arXiv preprint arXiv:2404.06654 (2024).
>
> [4] Bai, Y., Lv, X., Zhang, J., Lyu, H., Tang, J., Huang, Z., ... & Li, J. (2023). Longbench: A bilingual, multitask benchmark for long context understanding. arXiv preprint arXiv:2308.14508.
>
> [5] An, S., Ma, Z., Lin, Z., Zheng, N., & Lou, J. G. (2024). Make Your LLM Fully Utilize the Context. arXiv preprint arXiv:2404.16811.

---

> ### Comment · Reviewer_19vD · 2024-12-03
>
> Thanks for the response. I didn't get any notification from openreview about the update. Sorry for the late reply. All of my concerns are addressed and I raised the score.

---

> > ### Author Response · Authors · 2024-12-03
> >
> > Thank you for raising the score! We will incorporate our discussions into our next revision.

---

### Official Review · Reviewer_Vib9 · 2024-11-01

**Soundness:** 3
**Presentation:** 3
**Contribution:** 3
**Rating:** 6
**Confidence:** 3

**Summary:**

The paper targets the challenge of LLM accurately retrieving information and maintaining reasoning capabilities when processing long-context inputs. To this end, the author designs a synthetic dataset specifically for fine-tuning recent popular pretrained models. An interesting advantage of this dataset is it doesn't contain any factual information. Their evaluations on long context retrival and reasoning tasks show clear improvements. They also compare with three additional long-context augmentation datasets as baselines. Results show that their synthetic dataset achieves comparable performance in long-context retrieval and reasoning without causing the significant degradation on general benchmarks observed with other baselines.

**Strengths:**

The paper is well written and easy to follow.

The problem addressed in this paper is pretty meaningful.

The experiments are comprehensive and effectively support the main points.

The idea is straightforward but produces good results.

**Weaknesses:**

I have some concerns about the experiments.

(1) For stage 1, why does the author fine-tune Mistral 7B for 2 epochs but fine-tune GPT-3.5 Turbo for 3 epoches? Is the performance sensitive to the number of training epochs?

(2) For stage 1, does it need any training strategy such as early stopping? Will a longer training hurt the performance on general benchmarks? Also, what will happen if we train with a larger key-value retrieval dataset? Will the performance on long context retrival and reasoning tasks further improve, and might it negatively affect general benchmark performance?

(3) Will the performance further improve if also fine-tune Mistral 7B with Multi-subkey dictionary key-value retrieval dataset?

(4) When fine-tuning with MDQA, does it train with 20 documents and place a gold document in some positions? If so, does this mean that the model is fine-tuned with too few samples?

(5) The author presents fine-tuning with MDQA as a baseline in Section 3.2.1 but does not provide similar results for FLENQA in Section 3.2.2. Is there a specific reason for this? I’m curious whether the conclusions would remain consistent.

(6). How many tokens are used for training for the other baselines in stage 4? I think it's a little bit hard to say if this method beats the other baseline because the baselines actually shows good performance on long context retrival and reasoning tasks especially on FlenQA(cot), the performance is also ok on some datasets in the general benchmarks.

A minor suggestion: I think it would be helpful to include the average accuracy/gap across all datasets in Table 2, as it would clarify the overall average degradation.

**Questions:**

(1). Is there any reason the dataset is built with 3 or 4 digits? I think 4 digits will count as 2 tokens in the GPT tokenizer, while 3 digits is just 1.
(2) Can GSM8K be considered a reasoning task? I think Triviaqa can also be considered a retrieval and reasoning task. Why can't those datasets improve performance?

---

> ### Author Response · Authors · 2024-11-30
> **Response to Reviewer Vib9 (1/4)**
>
> Dear Reviewer Vib9,
>
> We would first like to apologize for our delayed reply. We greatly appreciate your constructive feedback on our paper. Below, we address the specific concerns raised in the review.
>
> **W1 & W2: The choice of epochs and whether the performance is sensitive to the number of training epochs.**
>
> Thanks for pointing this out. There is no specific reason on the number of epochs we choose. For GPT-3.5, since we use OpenAI's API and we don't know exactly how it works internally, we use the parameter `auto` when finetuning and the API automatically selects epoch to be 3. For Mistral, we choose epoch number to be 2 and find that the performance improves on MDQA and FLenQA, so we didn't test other epoch numbers.
>
> To address your concern on different amount of training, we conduct additional experiments in Appendix B.1 that study how the amount of training affect model's performance on MDQA, FLenQA and general capabilities benchmarks. In particular, we train Mistral 7B on the following settings
>
> 1. simple dictionary key-value retrieval with dataset size 350 and for 1 epochs, denoted as `sd (ep1)`
> 2. `sd (ep2)`, which is the setting we considered in Section 3
> 3. `sd (ep4)`
> 4. `sd x2 (ep2)`, where the training set doubled and train with 2 epochs
>
> We show our results on MDQA and FLenQA in [Figure 12](https://github.com/PlJQ/needles/blob/main/Figure_12.pdf) and observe that
>
> * Early stopping / undertraining (`sd (ep1)`) still improves from the original model but has a slight gap from other cases with more amount of training.
> * Further training on the same data (`sd (ep4)`) will have similar performance with `sd (ep2)` on MDQA and FLenQA (no cot) and have a slight improvement on FLenQA (cot).
> * Training with larger dataset for the same epoch (`sd x2 (ep2)`) will have slight improvement on FLenQA and slight degradation on MDQA compared to `sd (ep2)`
>
> The results of general benchmark evaluation is in [Table 3](https://github.com/PlJQ/needles/blob/main/Table_3.pdf). We observe no significant degradation (there is a slight degradation on GSM8K for `sd x2 (ep2)`). We also conduct the same experiment for MultidocQA and IN2. The results for MultidocQA are shown in [Figure 13](https://github.com/PlJQ/needles/blob/main/Figure_13.pdf) and [Table 4](https://github.com/PlJQ/needles/blob/main/Table_4.pdf); and the results for IN2 are shown in [Figure 14](https://github.com/PlJQ/needles/blob/main/Figure_14.pdf) and [Table 5]( https://github.com/PlJQ/needles/blob/main/Table_5.pdf). We can observe that while further training or larger dataset can also further improve the model's performance with MultidocQA or IN2 data, the degradation on general benchmark will be more significant.

---

> ### Author Response · Authors · 2024-11-30
> **Response to Reviewer Vib9 (2/4)**
>
> **W3: Will the performance further improve if also fine-tune Mistral 7B with Multi-subkey dictionary key-value retrieval dataset?**
>
> Inspired by your question, in Appendix B.2 of our revised draft (page 18-20), we conduct additional experiments to test whether finetuning Mistral on "harder tasks" (which we will introduce what we mean by this later) will further boost the performance. The conclusion is that directly finetuning Mistral 7B on "harder tasks" won't boost the performance, but the performance can increase if we first finetune Mistral on simple dictionary key-value retrieval (denoted as `sd`) and then finetune it on other "harder tasks".
>
> In particular, we consider two "harder tasks":
> 1. multi-subkey key-value retrieval, denoted as `msd`
> 2. simple dictionary key-value retrieval, denoted as `sdvar`, where the answer depends on multiple parts of the context. An example is shown in [Figure 15](https://github.com/PlJQ/needles/blob/main/Figure_15.pdf).
>
> We finetune Mistral directly on these two tasks and we also consider the cases where we first finetune on `sd` and then finetune on `msd` or `sdvar` to simulate an "easy-to-hard" learning process as `sd` is simpler than the other two. In particular, we train the model with the following settings:
> 1. train on `msd` for 2 epochs, denoted as `msd (ep2)`
> 2. train on `sd` for 2 epochs and then on `msd` for 2 epochs, denoted as `sd (ep2)->msd (ep2)`.
> 3. train on `sdvar` for two epochs, denoted as `sdvar (ep2)`,
> 4. train on `sd` for 2 epochs and then on `sdvar` for 2 epochs, denoted as `sd (ep2)->sdvar (ep2)`.
>
> We show the results in [Figure 16](https://github.com/PlJQ/needles/blob/main/Figure_16.pdf) and notice that while training Mistral just on `msd` or `sdvar` does not improve the performance (and the performance decreases on FlenQA no-cot), first train it on `sd` and then train it on `msd` or `sdvar` can improve model's performance on MDQA and FLenQA (cot) while maintaining the same performance in FLenQA (no-cot).
>
> We conduct additional experiment that trains `sd`, `msd` or `sdvar` for 4 epochs to see if the improvement was simply because we don't train enough, and the results in [Figure I](https://github.com/PlJQ/needles/blob/main/Figure_I.pdf) shows that `sd (ep2)->msd (ep2)` and `sd (ep2)->sdvar (ep2)` still have better performance compared to `sd (ep4)`, `msd (ep4)` and `sdvar (ep4)`, indicating that the training order here does help.
>
> We also show model's performance on general benchmark in [Table 6](https://github.com/PlJQ/needles/blob/main/Table_6.pdf), where we find a slight degradation on GSM8K with`sd (ep2)->msd (ep2)` and `sd (ep2)->sdvar (ep2)`. We hypothesize that this slight degradation is due to that we use integers as keys and values, which might slightly influence the model's understanding on numbers. A future direction is to add a set of special retrieval tokens and train the model on retrieval tasks using these tokens ([1] has similar idea of "landmark" token, but it requires explicit modification of the attention during inference time). On the other hand, [Table 7](https://github.com/PlJQ/needles/blob/main/Table_7.pdf) shows that the counterparts on IN2 (`IN2 (ep2)->IN2 (ep2)`, which is first train Mistral on IN2 for 2 epochs and then train it on new IN2 data for 2 epochs) and MultidocQA (`MultidocQA (ep2)->MultidocQA (ep2)`) suffer from more severe degradation.
>
> **W4: When fine-tuning with MDQA, does it train with 20 documents and place a gold document in some positions? If so, does this mean that the model is fine-tuned with too few samples?**
>
> Thanks for pointing this out. We apologize for causing the confusion here. Here we consider "20 Documents MDQA" as a task where each sample of this task contains 20 documents as a context and a question based on the context. When we finetune the model on "20 Documents MDQA", what we mean is that when we construct the finetuning dataset, we construct $n$ training samples ($n=150$ for GPT and $n=350$ for Mistral 7B, matching the corresponding size of our artificial dataset) where each sample is a "20 Documents MDQA" task. Therefore, the number of samples and the total number of training tokens are the same (as each sample has roughly 4K tokens) when we compare our method to finetuning on MDQA.
>
> We will include more explanation in our future revised manuscript to make this part clearer.

---

> ### Author Response · Authors · 2024-11-30
> **Response to Reviewer Vib9 (3/4)**
>
> **W5: Finetuning on FlenQA**
>
> The reason why we did not finetune on FLenQA is that FLenQA [2] is a dataset that only contains $300$ unique questions / context. For each unique question, [2] places each question under different context with different context sizes with different padding types and dispersion strategies to make the evaluation set summing to 12K testing samples. However, if we want to train on FLenQA, excluding the test examples, we will only have $250$ unique samples, which are not enough for forming a single dataset for Mistral (as we need $350$ samples). While this number is enough for forming a single dataset for GPT-3.5, it is still not enough for testing it with multiple runs (each run with different data) to make a conclusion that is robust.
>
> We thank the reviewer for raising this question and will explain this in our revised manuscript.
>
> **W6: Other baselines**
>
> > How many tokens are used for training for the other baselines in stage 4?
>
> For all comparisons, unless otherwise specified (e.g., in Appendix B where we use different training data size or training epochs), the number of training tokens are the same. For example, in [Figure 8](https://github.com/PlJQ/needles/blob/main/Figure_8.pdf), all baselines are trained on $350$ samples (each sample with roughly 4K context length) for $2$ epochs. We will make this clearer in our revised manuscript.
>
> > I think it's a little bit hard to say if this method beats the other baseline because the baselines actually shows good performance on long context retrieval and reasoning tasks especially on FlenQA(cot), the performance is also ok on some datasets in the general benchmarks.
>
> It is true that training with MultidocQA significantly improves the performance on MDQA and training with IN2 significantly improves the performance on FLenQA (cot). However we still think our work is valuable for the following reasons.
> * Both MultidocQA and IN2 require access to GPT-4 when constructing the answer, and we cannot guarantee if the answer is 100% correct. Furthermore, as mentioned in our paper, real data might contain outdated information (which requires re-constructing it periodically) while synthetic data do not face this issue.
> * MultidocQA is a dataset of multiple documents question and answering where the model needs to paraphrase the document before answering. Therefore it can seen as a further enhancement of MDQA data and performs well on MDQA task.
> * For IN2 since the answer is deduced from one or multiple parts of the context, the answer (from GPT-4) contains GPT4's chain-of-thought reasoning. Finetuning on such data is also distilling GPT-4's chain-of-thought reasoning capability. In [Figure 8](https://github.com/PlJQ/needles/blob/main/Figure_8.pdf) we can see that IN2 has a significant gap on FLenQA (cot) compared to other data but the gap on FLenQA (no cot) is not that significant.
> * In our response to [W1 & W2](https://openreview.net/forum?id=8m7p4k6Zeb&noteId=QLnVTKcOUC) where we discuss how different amount of training affect the model's capability, we observe that with further training or larger dataset, the degradation is more severe if we train the model on IN2 or MultidocQA data than if we train the model on our synthetic data ([Table3](https://github.com/PlJQ/needles/blob/main/Table_3.pdf), [Table 4](https://github.com/PlJQ/needles/blob/main/Table_4.pdf), [Table 5](https://github.com/PlJQ/needles/blob/main/Table_5.pdf)).
> * We acknowledge that synthetic data cannot fully replace real data. For example, if we want to train a long-context specialized model, training with real data might be the right choice as the real data enhances long-context capability better. On the other hand, synthetic data can still be valuable for generalist model as it enhances model's long-context capabilities on some tasks while having no (or little) degradation on other tasks.
> * At last, our study shows that for real tasks that require some particular skills, there are some corresponding synthetic tasks that can enhance those skills without severely degrading model's general capability (that real data might do). For example, simple dictionary key-value retrieval corresponds to single "key-value" retrieval capability; the simple dictionary key-value retrieval variant in [our response to W3](https://openreview.net/forum?id=8m7p4k6Zeb&noteId=CGyNOsURjL) correspond to a higher-level capability where the model needs to focus on different parts of the context to deduce the answer. We show the existence of such real $\leftrightarrow$ synthetic correspondence on long-context tasks (MDQA and FLenQA) and think the finding itself is valuable. We hope our work could highlight the importance of synthetic data.
>
>
> **Minor suggestion**
>
> Thank you for you suggestion. We will update our table in our next revision.

---

> ### Author Response · Authors · 2024-11-30
> **Response to Reviewer Vib9 (4/4)**
>
> **Q1: Is there any reason the dataset is built with 3 or 4 digits? I think 4 digits will count as 2 tokens in the GPT tokenizer, while 3 digits is just 1.**
>
> There is no specific reason why we choose these numbers. In our dataset a key/value has an equal probability of being 3 or 4 digits. It is true that GPT tokenizes 4 digits with 2 tokens and 3 digits with 1 token, but that doesn't make much difference because the model will output *gold value* with the same tokenization as in the prompt. If the *gold value* has 2 tokens, then the model will answer 2 tokens for the *gold value* part.
>
> **Q2: Can GSM8K be considered a reasoning task? I think Triviaqa can also be considered a retrieval and reasoning task. Why can't those datasets improve performance?**
>
> In this paper we mostly focus tackling the problems in the long-context setting. In particular
> 1. When important information is placed at different positions in the context under a long-context setting, model's capabilities drop when the position in the middle or at the end ("lost-in-the-middle" phenomenon and evaluated by MDQA)
> 2. When we put irrelevant information in the context, the model's reasoning performance drop (evaluated by FLenQA) as the context sizes grow.
>
> GSM8K is a short-context task so the performance won't be improved as our data (and the baselines) solve the issue that "reasoning capability degrades as the context length increases", which is measured by FLenQA. For TriviaQA, while it can be also think of as "retrieval", it is different from the retrieval task we consider where we provide an explicit context and ask the model to retrieve. Instead, it is more about retrieving information in model's internal "memory", so it is also beyond the retrieval capability we are trying to improve.
>
> We thank again for your thoughtful questions and comments. We would be happy to discuss any further questions.
>
> **References**
>
> [1] Mohtashami, A., & Jaggi, M. (2023). Landmark attention: Random-access infinite context length for transformers. arXiv preprint arXiv:2305.16300.
>
> [2] Levy, M., Jacoby, A., & Goldberg, Y. (2024). Same task, more tokens: the impact of input length on the reasoning performance of large language models. arXiv preprint arXiv:2402.14848.

---

> ### Comment · Reviewer_Vib9 · 2024-12-02
>
> Thank you for your response. I think most of my questions are addressed, but I still have some questions and concerns.
>
> Question: Does the answer to Q1 mean that the length and type of the gold value will not influence the performance?
>
> Concern 1: The answer to Q2 also makes me doubt the scope of applications of this work and agree with the W1 mentioned by Reviewer G5rG. I think this work only targets a narrow definition of retrieval and reasoning tasks, and even the real task experiments in the paper sound a little bit unrealistic to me. I think in the real world, the other documents may not be completely irrelevant. Also, as mentioned by the author, there might be some outdated information in real applications. I feel like the real task is just the longer version of the synthetic data, which decreases the contribution of this work. Could you please list some real applications to help me understand the contribution?
>
> Concern 2: The experiments show that while further training or larger dataset can also further improve the model's performance with MultidocQA or IN2 data, the degradation on general benchmark will be more significant. I think this confirms my concern.  Finetuning truly harms the general purpose capabilities of the models. While in real tasks, it always not only purely provides a long context and asks the model to retrieve, but also needs other general capabilities.

---

> ### Author Response · Authors · 2024-12-03
> **Follow-up reply (1/2)**
>
> Dear Reviewer Vib9,
>
> Thank you for your reply and we appreciate your thoughtful questions. Below we share our response.
>
> ## Question
>
> Thank you for raising this question. When generating each key / value, it has 0.5 chance of being a 3 digit number and 0.5 chance or being a 4 digit number. The tokenization will not cause a problem: If the *gold value* has one token, the model will answer one token, and so on for two or more tokens.
>
> However, changing the length of key / value can potentially change the difficulty of the task (which can change the model's performance). For example, if we fix the context size, having shorter keys / values will increase the total number of key-value pairs because we want to generate as many dictionaries & key-value pairs as possible to fit the context size. On the other hand, having longer keys / values will then decrease the number of key-value pairs, and it can be a more difficult task for the model if the model cannot retrieve long value (long sequence of tokens) or identify long keys. Changing the type of key / value (e.g., use random strings) can also possibly change the difficulty of the retrieval task, depending on how the model tokenizes the string. We think it can be an interesting direction for future work to finetune the model on tasks with different length and types of keys / values.
>
> We will incorporate our discussion into our next revision.
>
> ## Concern 1
>
> > I think this work only targets a narrow definition of retrieval and reasoning tasks
>
> We would like to clarify that the primary setting of this paper is "long-context" setting and we focus on long-context retrieval and long-context reasoning. Therefore, "retrieving internal knowledge" and "short-context reasoning" are orthogonal to our work. In this work, we fix two well-documented problems:
> 1. When provided with long context where important information is placed at different positions in the context, model's capability to accurately retrieval the information from the context and answer the question drops when the position in the middle or at the end ("lost-in-the-middle" phenomenon and evaluated by MDQA).
> 2. When provided with long context and where irrelevant information is in the context, the model's reasoning performance drops (evaluated by FLenQA).
>
> Since finetuining the model on our dataset will mitigate these problems with no significant degradation on general benchmarks, we think doing so is still valuable, compared to other data augmentation dataset where there is a more significant tradeoff on model's general capability.
>
> > I feel like the real task is just the longer version of the synthetic data, which decreases the contribution of this work.
>
> Thanks for raising this concern. In this paper, "synthetic" means more like "algorithmic" or "symbolic", where the task can be represented by symbols that has no real-world knowledge (e.g., integers) and solved by a demternistic algorithm; "real" task here refers to a task that needs some understanding of the context or some reasoning. For example,
> * A question in MDQA asks `What is the cross on a letter t called`. The answer is `crossbar` and the information in gold document that answers this question is
>   * `When the strokes connect as in A and H or cross strokes as in t is known as crossbar`
> * A question in FLenQA asks `Is Samantha Arnold younger than Julian Barton?` and the following information that are required to answer the question appear at different locations in the context.
>   * `Julie Baker is younger than Julian Barton.`
>   * `Samantha Arnold is younger than Julie Baker`
>
> We hope this better clarifies the notion of "synthetic" and "real" and we apologize for the confusion. We will incorporate our discussion into our next revision to clarify this.
>
> In addition, we think that the finding "finetuning LLMs on synthetic task improves the performance on real task" (in particular, Finding 1&3 in the paper) itself is valuable, indicating that the model can effectively transfer its learned capabilities. Thank you again for mentioning this. We will incorporate our discussion into our next revision.

---

> ### Author Response · Authors · 2024-12-03
> **Follow-up reply (2/2)**
>
> > I think in the real world, the other documents may not be completely irrelevant. Could you please list some real applications to help me understand the contribution?
>
> Thank you for raising this concern. In Section 4, we discussed a variant of MDQA [1] where all distractors are relevant distractors (other documents being distractors that talk about relevant topic in the question but does not answer the question, where the distractors are selected by a retrieval system using a relevance score) and observe that in this case finetuning the model on simple dictionary key-value retrieval (`sd`) will not improve significantly from the original model. We further investigate whether the performance will improve if we first finetune `sd` and then on multi-subkey dictionary key-value retrieval (`msd`), as there are also "distractors" in `msd` (keys that share some sub-keys with the gold key). In [Figure II](https://github.com/PlJQ/needles/blob/main/Figure_II.pdf) we observe that `sd->msd` can slightly increase the performance. On the other hand, finetuning the model on other baselines can sometimes degrade the performance. However, we think that this is a hard task and it does not truly capture the real-world scenario.
>
> **To address your concern, we conduct an additional experiment on another setting**: "20 Documents MDQA variant" where there is 1 gold document, 9 relevant distractors and 10 irrelevant (random) distractors. This better simulate the real setting where some (but not all) part of the context is relevant to the question. In [Figure III](https://github.com/PlJQ/needles/blob/main/Figure_III.pdf), we can observe that `sd (ep2)->msd (ep2)` has a better improvement over the original model. This shows that our synthetic dataset can still be useful in settings where other documents are not completely irrelevant. A possible real application is to input a long novel as context and ask an LLM a question about a character in the novel that appears several times (which corresponds to the setting where multiple documents are relevant to the question).
>
> > Also, as mentioned by the author, there might be some outdated information in real applications.
>
> We apologize for causing the confusion here. What we mean here by "outdated information" is that real long-context augmentation data like IN2 and MultidocQA contains real-work information, and some of the information can become outdated as the time goes on. For example, one training sample of MultidocQA provides a list of documents and asks `What year was the most recent Super Bowl held?` with answer `The year is 2023` (excluding the reasoning part) as the dataset was built in 2023 (while the true answer now should be 2024). Therefore, for real data like MultidocQA (IN2 also has similar cases), because the model is finetuned on the answer, beyond learning to retrieve the correct information, the model also learns the information about the real-world, and such information can be outdated and it requires re-building the dataset and re-finetuning on such dataset. On the other hand, our proposed dataset will not suffer from this issue.
>
> ## Concern 2
>
> > Finetuning truly harms the general purpose capabilities of the models. While in real tasks, it always not only purely provides a long context and asks the model to retrieve, but also needs other general capabilities.
>
> It could be true that some real tasks require multiple general capabilities. However, we think that here when finetuning on real long-context augmentation data, in particular IN2 & MultidocQA, the capability it hurts is model's capability to provide accurate information (in model's "internal memory"). This can be seen from [Table 4](https://github.com/PlJQ/needles/blob/main/Table_4.pdf) and [Table 5](https://github.com/PlJQ/needles/blob/main/Table_5.pdf) where the accuracies on TriviaQA and NQ-Open (two tasks that are knowledge-based question-answering tasks that tests model's internal knowledge) drop. We think that this is because finetuning the model on data that contains real world knowledge encourages hallucinations (e.g., the "Super Bowl" example above in our response to **Concern 1**) and recent work [2] also confirms this. On the other hand, synthetic data does not rely on real-world information and therefore will not suffer from the encouraging hallucination.
>
> Thank you once again for your constructive feedback. We are happy to address any remaining concerns you have.
>
> [1] Liu, N. F., Lin, K., Hewitt, J., Paranjape, A., Bevilacqua, M., Petroni, F., & Liang, P. (2024). Lost in the middle: How language models use long contexts. Transactions of the Association for Computational Linguistics, 12, 157-173.
>
> [2] Gekhman, Z., Yona, G., Aharoni, R., Eyal, M., Feder, A., Reichart, R., & Herzig, J. (2024). Does Fine-Tuning LLMs on New Knowledge Encourage Hallucinations?. arXiv preprint arXiv:2405.05904.

---

### Official Review · Reviewer_G5rG · 2024-11-02

**Soundness:** 2
**Presentation:** 3
**Contribution:** 2
**Rating:** 6
**Confidence:** 3

**Summary:**

The work proposes a set of synthetic tasks based dictionary-based key-value retrieval and uses it extend the context lengths of Mistral-7B-0.1 and GPT-3.5 turbo models. Finetuning on these datasets improves performance on MDQA and FLenQA benchmarks while discouraging hallucinations. Performance on general evaluation benchmarks like TruthfulQA, GSM8k is shown to be retained.

**Strengths:**

- Finetuning on randomly sampled key-value datasets is able to fix the “lost-in-the-middle” phenomenon for GPT-3.5 while retaining original knowledge.
- Finding that finetuning with answer templates performs better.
- Work on improving LLMs through synthetic datasets is crucial for the community.

**Weaknesses:**

1) The proposed synthetic dataset targets only retrieval tasks, like MDQA, while other important applications of long-context, such as in-context learning and RAG, involve \emph{understanding} the context as a whole. Hence, I have doubts about the scope of applications of this work. Concurrent works [1, 2] argue that finetuning/improving only retrieval capabilities does not capture all long-context applications.

2) In line 263, it is claimed that "..proposed data mitigates this primacy bias..", but the curve in Fig 5(b) still shows a descent, although with a higher accuracy.

3) About line 270, "..we also finetune the models on the MDQA dataset ..", from what I have understood, the difference between performance upon finetuning on the proposed dataset vs MDQA, can arise from two factors:
  (a) Randomization of the position of the gold document.
  (b) Complexity due to requirement of english understanding.
Since point (a) is easy to track, further study is needed to understand the differences between the two. Point (b), for example, can be explored by including the <Mistral-7B ft with MDQA> row in Table 1.

4) In Section 3.4, a possible reason MultiDocQA & IN2 outperform the proposed dataset is that the baselines require extracting information from multiple contexts. Can you solve this drop by adding a task that needs to retrieve the values of multiple keys? Such a task also discourages the hallucinations that MultiDocQA & IN2 seem to give rise to.

5) Regarding line 214, It is unclear why the finetuning and evaluation is fixed within one sliding window. Since some works can improve the maximum context length of Mistral (32k), evaluation at longer context lengths is needed. Section 3.5 is the right step towards this, but quality evaluations still need to be included.

6) Although manipulation of positional embeddings to extend context lengths without training is orthogonal to this work, Works like [3, 4] improve the context lengths of Mistral while maintaining performance on retrieval-like benchmarks. Either an explanation in related work or a comparison with baselines is needed.

Overall, I feel further studies and evaluations are required to improve the work. The work claims multiple interesting findings that require more comprehensive analysis and explorations. Happy to discuss more during rebuttal.

[1] Is It Really Long Context if All You Need Is Retrieval? Towards Genuinely Difficult Long Context NLP. CoRR abs/2407.00402 (2024)

[2] Michelangelo: Long Context Evaluations Beyond Haystacks via Latent Structure Queries. CoRR abs/2409.12640 (2024)

[3] LongEmbed: Extending Embedding Models for Long Context Retrieval. CoRR abs/2404.12096 (2024)

[4] LLM Maybe LongLM: SelfExtend LLM Context Window Without Tuning. ICML 2024

**Questions:**

<see Weaknesses>

---

> ### Author Response · Authors · 2024-11-30
> **Response to Reviewer G5rG (1/3)**
>
> Dear Reviewer G5rG,
>
> We would first like to apologize for our delayed reply. We greatly appreciate your thoughtful feedback on our paper. Below, we address the specific questions raised in the review. (We will address Weakness 1 in the end as our response to Weakness 1 relates to our responses to other concerns).
>
> **W2: Phrasing on "Primacy Bias"**
>
> We thank the reviewer for pointing this out. We rephrased that part in our revised draft (page 5). We acknowledge that the primacy bias is not fully mitigated, despite the fact that the descent in accuracy diminishes, especially in the case of finetuning with a template. However, we believe that the overall improvement is an interesting observation.
>
> **W3: Mistral ft on MDQA**
>
> The randomization is probably not causing the problem as we randomly generate random positions for each sample and the results are average across 3 runs where each run has 350 samples. We also added a row in Table 1 in our updated draft and it shows the finetuning on MDQA does degrade model's general capability.

---

> > ### Author Response · Authors · 2024-11-30
> > **References**
> >
> > **References**
> >
> > [1] Is It Really Long Context if All You Need Is Retrieval? Towards Genuinely Difficult Long Context NLP. CoRR abs/2407.00402 (2024)
> >
> > [2] Michelangelo: Long Context Evaluations Beyond Haystacks via Latent Structure Queries. CoRR abs/2409.12640 (2024)
> >
> > [3] LongEmbed: Extending Embedding Models for Long Context Retrieval. CoRR abs/2404.12096 (2024)
> >
> > [4] LLM Maybe LongLM: SelfExtend LLM Context Window Without Tuning. ICML 2024
> >
> > [5] Mohtashami, A., & Jaggi, M. (2023). Landmark attention: Random-access infinite context length for transformers. arXiv preprint arXiv:2305.16300.

---

> ### Author Response · Authors · 2024-11-30
> **Response to Reviewer G5rG (2/3)**
>
> **W4: A possible reason MultiDocQA & IN2 outperform the proposed dataset is that the baselines involve extracting information from multiple contexts. Can you solve this drop by adding a task that needs to retrieve the values of multiple keys?**
>
> Thanks for your suggestion. Firstly we think that a task to retrieve values of multiple keys will reduce to multiple consecutive simple dictionary key-value retrieval tasks (denoted as `sd`). However, inspired by your suggestion, in Appendix B.2 of our revised draft (page 18-20), we conduct additional experiments to test whether finetuning Mistral on "harder tasks" (which we will introduce what we mean by this later) will further boost the performance. The conclusion is that directly finetuning Mistral 7B on "harder tasks" won't boost the performance, but the performance can increase if we first finetune Mistral on `sd` and then finetune it on other "harder tasks".
>
> We design a new task called simple dictionary key-value retrieval variant (denoted as `sdvar`), where multiple *gold values* are associated with the *gold key* and we ask the model to report all *gold values* in ascending order of values (example shown in [Figure 15](https://github.com/PlJQ/needles/blob/main/Figure_15.pdf). This is a task where the answer depends on multiple parts of the context. We also consider multi-subkey dictionary key-value retrieval (denoted as `msd`) because it requires some awareness of other parts of the context as other keys share some subkeys with the *gold key*.
>
> We finetune Mistral directly on these two tasks and also consider the cases where we first finetune on `sd` and then finetune on `msd` or `sdvar` to simulate an "easy-to-hard" learning process as `sd` is simpler than the other two. In particular, we train the model with the following settings:
> 1. train on `msd` for 2 epochs, denoted as `msd (ep2)`
> 2. train on `sd` for 2 epochs and then on `msd` for 2 epochs, denoted as `sd (ep2)->msd (ep2)`.
> 3. `sdvar (ep2)`,
> 4. `sd (ep2)->sdvar (ep2)`.
>
> We show the results in [Figure 16]( https://github.com/PlJQ/needles/blob/main/Figure_16.pdf) (here we do not show the error bars as they will be intersecting with each other; the results are averaged) and notice that while training Mistral just on `msd` or `sdvar` degrade the performance, first train it on `sd` and then train it on `msd` or `sdvar` can improve model's performance on MDQA and FLenQA (cot) while maintaining the same performance in FLenQA (no-cot). We also conduct additional experiment that trains `sd`, `msd` or `sdvar` for 4 epochs to see if the improvement was simply because we don't train enough, and the results in [Figure I](https://github.com/PlJQ/needles/blob/main/Figure_I.pdf) shows that `sd (ep2)->msd (ep2)` and `sd (ep2)->sdvar (ep2)` still have better performance compared to `sd (ep4)`, `msd (ep4)` and `sdvar (ep4)`, indicating that the training order here does help.
>
> We then compare `sd (ep2)->msd (ep2)` and `sd (ep2)->sdvar (ep2)` with four baselines:
> 1. `IN2 (ep2)`
> 2. `IN2 (ep2)->IN2 (ep2)`
> 3. `MultidocQA (ep2)`
> 4. `MultidocQA (ep2)->MultidocQA (ep2)`
>
> In `IN2 (ep2)->IN2 (ep2)`, we first train on `IN2` (dataset size 350, matching `sd`, `msd` or `sdvar`) for 2 epochs followed by a new training set on `IN2` data for 2 epochs. In [Figure 17](https://github.com/PlJQ/needles/blob/main/Figure_17.pdf) we show the result can observe that
> * On MDQA, while `sd (ep2)->msd (ep2)` and `sd (ep2)->sdvar (ep2)` improve from `sd (ep2)`, there is still a gap with `MultidocQA` settings. A possible reason is that `MultidocQA` is a stronger augmentation dataset that consider tasks about multi-document question answering where the document is first paraphrased before being answered. Our dataset is still better than IN2 on MDQA.
> * On FLenQA (cot), the performance between `sd (ep2)->sdvar (ep2)` and `MultidocQA (ep2)->MultidocQA (ep2)` are close. However, while `sd (ep2)-sdvar (ep2)` improves from `sd (ep2)`, there is still a large gap with `IN2` or `IN2 (ep2)->IN2 (ep2)`. A possible reason why `IN2` performs so well on FLenQA (cot) is that when constructing the dataset, IN2 includes GPT-4 answer on question-answering, which involves chain-of-thought reasoning. In other words, `IN2` distilled some of GPT-4's chain-of-thought reasoning capability that might cause improved performance.
> * On FLenQA (no-cot), the gap between our dataset and `IN2` is small.

---

> ### Author Response · Authors · 2024-11-30
> **Response to Reviewer G5rG (3/3)**
>
> We also evaluate model's performance on general benchmarks (shown in [Table 7](https://github.com/PlJQ/needles/blob/main/Table_7.pdf)) and observe that the model finetuned on baseline methods suffer from hallucination, indicated by the degradation in knowledge-based evaluation like TrviaQA and NQ-Open; `MultidocQA (ep2)->MultidocQA (ep2)` also suffer from greater degradation on GSM8K. On the other hand, there is a slight degradation on GSM8K with`sd (ep2)->msd (ep2)` and `sd (ep2)->sdvar (ep2)`. We hypothesize that this slight degradation is due to that we use integers as keys and values, which might slightly influence the model's understanding on numbers. A future direction is to add a set of special retrieval tokens and train the model on retrieval tasks using these tokens ([5] has similar idea of "landmark" token, but it requires explicit modification of the attention during inference time).
>
> **W5: More evaluation in the longer context setting**
>
> Thanks for pointing this out and we apologize for causing the confusion. The reason why we consider the 4K window is that the original "lost-in-the-middle" primarily considers the 4K window setting and FLenQA is a eval set with maximum context size 3K. [Table I](https://github.com/PlJQ/needles/blob/main/Table_I.pdf) shows that model finetuned on `sd` on longer context (24K) will not cause significant degradation. We will incorporate this in our revised manuscript.
>
> **W6: Discussion of other related works on context extension**
>
> Thank you for mentioning. We don't choose works that manipulate positional encodings to extend the context window as baselines is because we consider different settings:
> * When running LLMs inference that exceeds the context window the model is trained on, LLMs will face the out-of-distribution (OOD) issue on positional encoding (that the model was not trained on during pretraining). Works like [4] change positional encoding that maps OOD positional encodings to in-distribution positional encodings so that the new positional encodings are within the range of pretraining context window.
> * In this work we consider a different setting where MDQA and FLenQA show that model still suffer a degradation on long-context tasks even if the task is within the context-window during pretraining.
> * In addition, [3]'s setting is on embedding models while we consider LLMs.
>
> We thank again for raising this concern and we plan to include more discussion of these related works in our next revised manuscript.
>
> **W1: Retrieval capability does not capture all long-context capabilities**
>
> Thank you for raising this concern. We agree that retrieval capabilities do not capture all long-context capabilities ([1, 2]) and we think synthetic data will not solve all long-context problems. However, we still think that our work with artificial synthetic dataset will contribute to the community for the following reasons:
> * We fix two well-documented problems: (1) when important information is placed at different positions in the context, model's capabilities drop when the position in the middle or at the end ("lost-in-the-middle" phenomenon and evaluated by MDQA); (2) when we put irrelevant information in the context, the model's reasoning performance drops (evaluated by FLenQA).
>     * Since finetuning the model on our dataset will mitigate these problems with no degradation on general benchmarks, we think doing so is still valuable (compared to other data augmentation dataset where there is a trade-off on general capabilities).
> * Our study shows that for real tasks that require some particular skills, there are some corresponding synthetic tasks that can enhance those skills without severely degrading model's general capability (that real data might do). For example, simple dictionary key-value retrieval corresponds to single "key-value" retrieval capability; the simple dictionary key-value retrieval variant in [our response to W4](https://openreview.net/forum?id=8m7p4k6Zeb&noteId=fXpQrgNuPS) correspond to a higher-level capability where the model needs to focus on different parts of the context to deduct the answer. We show the existence of such real $\leftrightarrow$ synthetic correspondence on long-context tasks (MDQA and FLenQA) and think the finding itself is valuable. We hope our work could highlight the importance of synthetic data.
>
> We thank again for your thoughtful questions and we would happy to answer other concerns.

---

### Meta-Review · Area_Chair_R8D4 · 2024-12-21

**Metareview:**

This paper proposes to mitigate "lost in the middle" on multi-document QA tasks via training the model on synthetic key-value retrieval data. Interestingly, the proposed method not only improves the accuracy when the ground truth documents are in the middle, but also avoids increasing hallucination compared with finetuning on the QA data directly. This opens door to a new direction for improving performance of RAGs while avoiding factuality tradeoffs via finetuning on non factuality related data. I tend to accept the paper if the conclusions stands in practical RAG systems. Despite recommending accept, I still have two uncertainties: (1) for the experiments supporting finding 6, are they trained with the same number of iterations? In Table 2, is it possible that you trained for more iterations on other tasks and caused forgetting? (2) How does the efficacy look like in practical retrieval systems where the retrieved documents are relevant in most cases?

**Additional Comments On Reviewer Discussion:**

In general I agree with the reviewers that this paper lacks experiments in practical RAG systems and this can be a weakness, but the idea and findings are interesting.

---

### Decision · Program_Chairs · 2025-01-22

Accept (Poster)